# Alzheimer’s Disease Early Diagnosis Using Manifold-Based Semi-Supervised Learning

**DOI:** 10.3390/brainsci7080109

**Published:** 2017-08-20

**Authors:** Moein Khajehnejad, Forough Habibollahi Saatlou, Hoda Mohammadzade

**Affiliations:** Department of Electrical Engineering, Sharif University of Technology, Azadi Avenue, Tehran 145888-9694, Iran; khajenejad_moein@ee.sharif.edu (M.K.); habibollahi_f@ee.sharif.edu (F.H.S.)

**Keywords:** Alzheimer’s disease, early diagnosis, semi-supervised manifold learning, label propagation, voxel-based morphometry, medical image analysis, image classification

## Abstract

Alzheimer’s disease (AD) is currently ranked as the sixth leading cause of death in the United States and recent estimates indicate that the disorder may rank third, just behind heart disease and cancer, as a cause of death for older people. Clearly, predicting this disease in the early stages and preventing it from progressing is of great importance. The diagnosis of Alzheimer’s disease (AD) requires a variety of medical tests, which leads to huge amounts of multivariate heterogeneous data. It can be difficult and exhausting to manually compare, visualize, and analyze this data due to the heterogeneous nature of medical tests; therefore, an efficient approach for accurate prediction of the condition of the brain through the classification of magnetic resonance imaging (MRI) images is greatly beneficial and yet very challenging. In this paper, a novel approach is proposed for the diagnosis of very early stages of AD through an efficient classification of brain MRI images, which uses label propagation in a manifold-based semi-supervised learning framework. We first apply voxel morphometry analysis to extract some of the most critical AD-related features of brain images from the original MRI volumes and also gray matter (GM) segmentation volumes. The features must capture the most discriminative properties that vary between a healthy and Alzheimer-affected brain. Next, we perform a principal component analysis (PCA)-based dimension reduction on the extracted features for faster yet sufficiently accurate analysis. To make the best use of the captured features, we present a hybrid manifold learning framework which embeds the feature vectors in a subspace. Next, using a small set of labeled training data, we apply a label propagation method in the created manifold space to predict the labels of the remaining images and classify them in the two groups of mild Alzheimer’s and normal condition (MCI/NC). The accuracy of the classification using the proposed method is 93.86% for the Open Access Series of Imaging Studies (OASIS) database of MRI brain images, providing, compared to the best existing methods, a 3% lower error rate.

## 1. Introduction

Alzheimer’s is a progressive disease where dementia symptoms gradually worsen over time. It destroys brain cells over time, causing memory and thinking skill losses. In early stages, also known as mild cognitive impairment (MCI), memory loss is mild, but with late-stage Alzheimer’s, the patient loses the ability to even carry on a conversation and respond to their environment. Alzheimer’s is the sixth leading cause of death in the United States. The estimated number of affected people will double for the next two decades so that one out of 85 persons will have Alzheimers disease (AD) by 2050 [1]. Those with Alzheimer’s live an average of eight years after their symptoms become noticeable. Although the greatest known risk factor for Alzheimer’s disease is aging and the majority of the patients are 65 and older, Alzheimer’s is not just a disease of old age. Up to 5 percent of people with the disease have early-onset Alzheimer’s, which often appears when someone is in their 40s or 50s. In Alzheimer’s disease, the hippocampus and cerebral cortex shrink while the ventricles enlarge in the brain. If the patient is in advanced stages of AD, these effects can be recognized in magnetic resonance imaging (MRI) images rather easily, though in the early stages it is a challenging task, with a high risk of a wrong prediction of the patient’s condition. Moreover, some of the symptoms found in the AD imaging data are also captured in imaging data of healthy aging people (age ≥75). Therefore, identifying the visual distinction between brain MRI images of older subjects with normal aging effects and those affected by AD, especially in mild stages, requires extensive knowledge and expertise. The diagnosis of Alzheimer’s disease requires a variety of medical tests which leads to huge amounts of multivariate heterogeneous data. It can be difficult and exhausting to manually compare, visualize, and analyze this data due to the heterogeneous nature of medical tests. Therefore, an efficient approach for accurate prediction of the condition of the brain through the classification of MRI images is greatly beneficial and yet very challenging. Additionally, in most cases, diagnosis based on MRI images must later be combined with additional clinical results for reliable classification of data. The reason that early diagnosis of AD is of great importance is that the clinical therapies given to patients are much more effective in slowing down disease progression and helping preserve some cognitive functions of the brain if the patients are in the early stages of their disease.

When relying on clinical evaluations which are based on cognitive measures, low sensitivity and specificity scores are obtained in early diagnosis of AD most of the time. Hence, in recent years some computer-aided approaches have been developed for low-cost, faster and more accurate diagnosis of AD. Various machine learning methods have been developed to predict AD. In previous works [2,3], deep learning was applied to capture high-level latent features from the images. The extracted features are later used for AD/MCI classification or just AD/normal condition (NC) classification in the method introduced by Sarraf et al. [4]. Furthermore, in a previously proposed method [5], a deep learning structure is used to extract features containing supplementary information and then a zero-masking strategy for data fusion is performed on multiple data modalities for this cause. To continue with this trend and in order to improve classical applications of deep learning, another previous effort [6] used the dropout technique. In another group of studies [7,8] linear support vector machines (SVM) are used for AD/NC classification of MRI images. Also, more recently a deep three-dimensional (3D) convolutional neural network was applied [9,10] to predict AD in its early or severe stages.

In this paper, we first start by selecting some of the most critical and drastic AD-related features using voxel-based morphometry (VBM) [11]. In order to discover voxel clusters which aid us to distinguish between AD patients and healthy subjects, Statistical Parametric Mapping software (SPM8) [12], was used to compute VBM. The dataset that we have used consists of two groups of subjects: (1) normal condition; and (2) subjects who were diagnosed with very mild to mild AD, all of whom were aged between 65 and 96 years old. The purpose of this work is to accurately distinguish between these two groups of subjects whose brain images are visually very similar in some cases. In the proposed MCI/NC classification method, after extracting a number of most informative features and for a faster and more efficient method, principal component analysis (PCA) [13] is performed to exploit an even more specific and effective subset of features that will help the computer get a more clear vision of the differences we are looking for between the two classes of subjects. Next, we continue by performing semi-supervised learning of the captured features. Finally, we carry out label propagation [14,15] from our training data to the rest of the dataset for an accurate prediction of the unknown labels.

Diagnosis of very early AD progression is intended to aid both researchers and clinicians to develop or test new treatments and monitor their effectiveness more easily. It is stated that AD pathologies could be detected in MRI images up to 3 years earlier than the actual clinical diagnosis [16]. Therefore, a machine learning method can be of great benefit for helping physicians make an accurate early diagnosis. On the other hand, the expected increasing costs of caring for AD patients, the workload of radiologists, and the limited number of available radiologists further demonstrate the necessity of having a computer-aided system for early, fast, and precise diagnosis and also for improving quantitative evaluations [17,18]. Furthermore, all previous efforts in the field as well as in the present study, when directed into a computer system, can be used as a second opinion by a physician to either verify their own diagnosis and increase its reliability or improve their final decision by getting help from the computer output in cases when they are less confident about their own diagnosis. Moreover, the possibility and benefits of practical usage of computer-aided diagnosis in clinical situations have been also the subject of a number of studies [19]. For instance, the radiologists’ performance while detecting clustered microcalcifications, which are small calcium deposits in breast soft tissue, both with and without the computer output has been observed and compared in one of these studies. It was proven in this study that the radiologists’ performance was improved significantly when computer output was also available. As a result of these studies, computer-aided diagnosis has recently become an important part of the routine clinical process for breast cancer detection in mammograms in the United States [20].

## 2. Theoretical Backgrounds

In the following sections we discuss the background relevant to this work. First of all we use G=(V,E) to denote a graph, where $V=(v_{1},v_{2},\cdots ,v_{N})$ is the set of nodes and $E=\left\{e_{i,j}\right\}$ is the set of edges. The edge $e_{i,j}$ indicates a connection between two nodes $v_{i}$ and $v_{j}$.

The adjacency matrix for a weighted graph is defined as a matrix A where ${\left[A\right]}_{ij}=w_{ij}$ if and only if nodes $v_{i}$ and $v_{j}$ are connected by an edge with weight $w_{ij}$ and ${\left[A\right]}_{ij}=0$ if they are not connected by an edge. The degree of a node $v_{i}$, denoted by $d(v_{i})$, is:
(1)$$d\left(v_{i}\right)=\sum_{j}{\left[A\right]}_{ij}$$
and the degree matrix D is defined as the following diagonal matrix, where the *i*-th diagonal element is $d(v_{i})$:
(2)$$D=\mathrm{diag}[d\left(v_{1}\right),d\left(v_{2}\right),\cdots ,d\left(v_{N}\right)]$$

### 2.1. Random Walk on a Graph

Random walk has been a subject of intensive study in the past decades and has been found useful in solving problems such as ranking [21], clustering [22,23], modeling diffusion processes [24,25] and synchronization [26,27]. Today it has become an important class of probabilistic models. In this section, we will briefly explain how a random walker navigates on a graph.

In a random walk, the walker currently at node *v* can move from *v* to any of its neighbouring nodes with a probability proportional to the weight of the edge between them.

The probability of the walker stepping into node $v_{j}$ from $v_{i}$ is denoted by $P(v_{j}|v_{i})$. Therefore, the stochastic process of the random walk is characterized by this transition matrix P. Each element of P follows the following equation:
(3)$${\left[P\right]}_{ij}=\frac{{\left[A\right]}_{ij}}{{\left[D\right]}_{ii}}=P\left(v_{j}|v_{i}\right)$$
where A is the adjacency matrix and D the degree matrix defined in the previous section. Hence, P can be written as:
(4)$$P=D^{-1}A$$

Let $P^{t}$ be the *t*-th power of P. Then, ${\left[P^{t}\right]}_{ij}$ represents the probability of the walker to arrive at node $v_{j}$ after exactly *t* steps, starting from node $v_{i}$.

### 2.2. Semi-Supervised Learning

Machine learning is a type of artificial intelligence that gives computers the ability to learn without being explicitly programmed. Evolved from the study of pattern recognition and computational learning theory in artificial intelligence, machine learning explores the study and construction of algorithms that can learn from and make predictions on data [28]. Utilizing machine learning, computer programs can be developed that can change when exposed to new unknown data. Machine learning uses that data to detect patterns in data and adjust program actions accordingly. From one perspective machine learning problems are categorized as being supervised, semi-supervised, or unsupervised. Here we want to briefly introduce semi-supervised learning.

In a semi-supervised method, feature vectors from unlabeled data are also used in the learning process in addition to the labels and feature vectors from the labeled ones. The information extracted from these unlabeled data will be beneficial for determining an approximation of the dispersion of data in the feature space. Before performing a semi-supervised learning algorithm, we need to make one important assumption:
if two members of the dataset are located in a dense region and are close to each other in the feature space, their labels will also be close to each other.

In this work, our goal is to label data with maximum accuracy knowing the labels of only a small number of images. We should acknowledge that, especially for a rather large dataset, labeling these images manually can be a tedious and difficult job. In particular, in mild stages, this diagnosis requires high-level proficiency. Therefore, it can now be understood why we have chosen to use a semi-supervised algorithm and how beneficial and also necessary a computer-based precise diagnosis can be.

### 2.3. Manifold Learning

Manifold learning [29] has always been of great interest for utilizing latent structural information from a dataset in a semi-supervised learning approach.

A manifold is a topological space that locally resembles Euclidean space near each point. A *k*-dimensional manifold in an *m* dimensional space is a surface in that space, such that for each point on this manifold, there exists a radial neighborhood consisting of a set of points on the manifold which have the following property: they can be mapped to a closed region in a *k*-dimensional linear space using a diffeomorphism, which is an invertible smooth function with a smooth inverse, that maps one differentiable manifold to another.

When applying manifold-based approaches to a specific learning problem, a dataset which is commonly expressed in an *m* dimensional space is indeed located in a non-linear subspace, or more specifically, on a *k*-dimensional manifold where k≪m.

Next, we are going to discuss two basic assumptions that we will completely fulfill as we go on.

Considering the fundamental assumption mentioned in the previous section, in a semi-supervised algorithm similar to the one we are aiming to apply to our problem, we will need to compute the distance between different data. Noticing that the data are now located on a manifold, it can be explicitly recognized that for a more effective result, rather than computing the Euclidean distance, we will need to define the forenamed distance on the manifold itself. This means calculating the geodesic distance which is the number of edges in the shortest path connecting them.Since in machine learning problems, we often possess only a limited number of training and test data, it is usually not possible to solve the manifold equation precisely. As a result, a graph is built up of an existing dataset as an approximation for the original manifold. After this graph is formed, considering *k*-nearest neighbor graphs corresponding to each node, we can assume that the Euclidean distance between two nodes connected with an edge approximately equals their geodesic distance. Also, regarding nodes which are not directly connected with an edge, the length of the minimum distance between them in the graph is a fair approximation of their geodesic distance.Moreover, keeping in mind that the fundamental assumption about semi-supervised algorithms also applies on this manifold, it can easily be concluded that the items of data which are located in dense areas on the manifold have similar labels. This implies that if a path exists between two members of the dataset which completely passes through the most probable and dense regions of the manifold, they will certainly have very close labels.Therefore, when using a graph as an approximation for such a manifold, it needs to have properties that also meet the above condition.

Manifold learning can be employed in various fields such as clustering, labeling, and also dimension reduction [30]. In this effort, our main purpose is to label data using a semi-supervised manifold learning. However, this specific method of labeling also requires an adequate dimension reduction which keeps the important latent structural information from all data while reducing very large dimensions to a convenient size.

### 2.4. Labeling Based on Manifold Learning

Let us assume we have a set of data with size *N* consisting of $v_{1},v_{2},\cdots ,v_{N}$ which belong to *c* different classes and we have been given the labels for the first *l* members of this set. This means that we know exactly what classes these *l* members belong to. We denote these labels with $y_{1},y_{2},\cdots ,y_{l}$. Our goal is to accurately find the labels for the rest of the dataset. In a manifold-based approach, to solve a classification problem with *c* different classes, we break it into *c* distinctive two-class problems in such a way that in each one of them the labeled data of a specific class have the label +1 where the rest of labeled data belonging to any of the other classes are labeled with –1. Therefore, what we are facing here is again a classification problem with just two classes. There are two different approaches to solving this type of classification problems. In the first method, a regression problem is defined where each item of unlabeled data is appointed a real number. These numbers are then compared to each other for each item of data in all *c* defined classification problems. Eventually, that specific item of data is given the label of the class that it has been assigned the largest number of times in the problem related to that specific class. In the second approach, according to the probabilities of each item of data belonging to each class in the corresponding defined classification problem, each unlabeled item of data will eventually belong to the class where it had been assigned the highest value probability.

Both these approaches must comply with all the manifold-related conditions and assumptions which were mentioned in the previous sections. Thus, in all these methods, the weights on the edges in the corresponding created graph, which we call *G*, are an appropriate function of the Euclidean distance between nodes as expressed below:
(5)$${\left[A\right]}_{ij}=w_{ij}=\left\{\begin{array}{cc} e^{\frac{{-\left|\right|v_{i}-v_{j}\left|\right|}^{2}}{2\sigma^{2}}} & i=j\mathrm{or}v_{i}⟷v_{j}\\ 0 & \mathrm{otherwise} \end{array}\right.$$
where *A* is the corresponding adjacency matrix of graph *G*, $v_{i}$ and $v_{j}$ are two arbitrary nodes in this graph, and $v_{i}⟷v_{j}$ indicates that $v_{i}$ and $v_{j}$ are connected with an edge. σ is the tuning parameter which will be set efficiently using cross validation. This procedure will be further discussed in the following sections. Here, we will briefly introduce a group of labeling methods based on random walks.

#### Random Walk-Based Labeling Approaches

In this section, we present a category of labeling methods which mainly rely on the second assumption made in Section 2.3. This assumption illustrates that if a pair of nodes is located in a dense region of the manifold and are close to each other, there is a high possibility of reaching the second node in a short time, starting a random walk from the first one. Based on this fact, a class of label propagation [14] methods has been developed, which can be explained in more detail as follows. In the first step, each labeled item of data acting as a labeled node in the graph has its own label with a weight which equals 1. Next, in each step, the labeled nodes distribute their labels among all their neighbors, giving their label to each neighboring node with a weight equal to the normalized weight of the edge between them. At the end of each step, the primarily labeled data gain back their own original label while the unlabeled data now have new sets of labels on them for continuing the process. This iterative procedure goes on until reaching a stationary state for the labels on all the nodes.

## 3. Methods and Materials

### 3.1. Dataset

The Open Access Series of Imaging Studies, OASIS (http://www.oasis-brains.org/app/template/Index.vm) [31], is a series of magnetic resonance imaging datasets from 416 subjects aged between 18 and 96 years, and includes a cross-section of the studied population. One hundred of the included subjects older than 60 years have been clinically diagnosed with very mild to moderate Alzheimer’s disease. The subjects are from both genders and are all right-handed. A rigid imaging protocol is strictly followed in the OASIS database in order to avoid any problems due to protocol variations while performing image normalization. Using a 1.5-T Vision scanner, in just a single imaging session, three to four T1-weighted magnetization-prepared rapid gradient echo (MP-RAGE) images were captured from every subject. In this study, we will exploit the averaged MP-RAGE image for each subject which is obtained through registration. First, for minimizing the variance between the first MP-RAGE image and the atlas target, which has been described in detail by Marcus et al. [31], a 12-parameter affine transformation was computed. Then a single, high-contrast, averaged MP-RAGE image was produced in atlas space by registering the remaining MP-RAGE images to the first one (in-plane stretch allowed) and resampling via transform composition into a 1-mm isotropic image in atlas space. This process is also discussed in more detail previously [31]. For gray–white contrast, MP-RAGE parameters were then optimized in several trials. The MRI acquisition details are reported in Table 1.

For this study, similar to the choice of other previous efforts [32,33,34], we selected 98 subjects with complete demographic, clinical or derived anatomic volume information, 49 of whom were diagnosed with very mild to mild AD, and the other half are healthy subjects. The additional information on the subjects is provided in Table 2. We have also reported the CDR score in the table. The CDR is a dementia staging instrument which gives ratings to each subject for impairment in each of the following six categories: memory, orientation, judgment, and problem-solving, function in community affairs, home and hobbies, and personal care. The global CDR is derived from individual ratings in each category. A global CDR equal to 0 means no dementia and numbers 0.5, 1,2 and 3 represent very mild, mild, moderate and severe stages, respectively.

As our future work, we aim to apply our method to another well-known data base: the Alzheimer’s Disease Neuroimaging Initiative (ADNI) (www.loni.ucla.edu/ADNI) as well.

### 3.2. Method

#### 3.2.1. Summary of the Method

Here, we will have an overlook on the main steps of our method. In this paper, we propose a novel approach for MCI/NC classification as the most crucial and beneficial type of classifier in AD diagnosis. We use a semi-supervised learning method for this goal. After extracting feature vectors containing high-level information using a method based on VBM, we attempt to conduct a label propagation method on a graph which is built as an approximation of these high-dimensional feature vectors. Figure 1 illustrates the different steps of the proposed method. In the following sections, we will completely discuss the introduced approach in detail.

#### 3.2.2. Image Processing and Feature Extraction

The process of extracting and then selecting high-level features that contain the most latent and crucial information, which can properly feed an accurate classifier, is an essential step that requires attention. Low-level or primitive features of an image are actually the visual content of the image which can be easily captured. These visual features include color and shape. On the other hand, there are high-level and latent features which are mostly texture-based and not very simple to capture. These are the features we are most interested in for the present work. The texture can be characterized by structure (spatial relationship) and also tone (intensity property).

#### 3.2.3. Voxel-based Morphometry (VBM)

Morphometry analysis has become a strong tool for carrying out quantitative measurements of the form and structural differences throughout the entire brain. Voxel-based morphometry (VBM) is a computational approach which performs a comparison on voxels of different brain images and then quantifies differences between local concentrations of brain images [35]. Recently, VBM has been applied in various studies in different fields. For instance, it can be used to perform a thorough study on the volumetric atrophy of the gray matter (GM) that exists in areas of neocortex in the brain and can be used to discriminate AD patients from healthy subjects [36,37].

Inspired by the method proposed in [11], we start the feature extraction process. This procedure includes four major phases. The first step requires the spatial normalization of all images before any further analysis is carried out. Now that all images are placed in a standard space, in the second phase, tissue classes are segmented using a priori probability map. Next, in order to perform smoothing via correcting any disruptive noise or small variations, the extracted information is convolved with a Gaussian kernel. The full width at half maximum (FWHM) of the applied Gaussian is set for any arbitrary problem accordingly. Finally, the last step is the voxel-wise statistical tests. In this phase, to express our data in terms of experimental and confounding effects and residual variability, the general linear model (GLM) [38] is utilized. Eventually, in order to build a statistical parametric map (SPM) [39], we need the computed contrast which is given by the GLM estimated regression parameters. The map is then thresholded according to the random field theory [40,41]. Figure 2 illustrates the different steps for performing VBM analysis.

#### 3.2.4. Image Processing and VBM in the OASIS Database

In this study, we aim to perform VBM as a method for investigating neuroanatomical differences in vivo. We exploit the average MRI volume reported for each subject in the OASIS dataset. Our goal is to benefit from the VBM method to obtain the proper spatial masks that we need for capturing the classification features. Here, we are specifically interested in GM and the information which lies in this tissue because experimental research suggests that the network within the gray matter, which is responsible for many of the higher order functions in the brain, is much more vulnerable to Alzheimer’s disease. This leads us to perform the VBM analysis on GM to distinguish between the regional concentration of GM among different subjects while ignoring global brain shape differences. We apply Statistical Parametric Mapping software (SPM8) [12], for this purpose which works in a right-handed coordinate system and therefore while pre-processing our data, we reorient all images to such a system. To start the process, we first need to notice that, as reported in previous effort in detail [31], all the images in this dataset are already registered and also re-sampled to 1-mm isotropic resolution in the target atlas space which has been biased already. Hence, no further spatial normalization will be needed. In the next step, tissue segmentation is achieved by combining probability maps and mixture model cluster analysis. No bias correction is required while performing tissue segmentation. As the last step, a spatial smoothing is essential before any statistical analysis is performed on voxels. A Gaussian kernel is applied at this point and the FWHM is manually set to 10 mm isotropic as suggested in past studies [11]. Smoothing is done mainly for increasing the signal to noise ratio and making up for any probable data loss that might have occurred while performing spatial normalization.

Now to create a GM mask, we compute the average of GM segmentation volumes from all subjects. The average GM segmentation is thresholded to obtain a binary mask including the voxels which have a probability greater than 0.1 in the average GM segmentation volume. Although the interpretation is not completely true due to the previously performed modulation, it is sufficiently accurate. Eventually, SPM8 employs GLM and carries out the required independent statistical tests to extract statistical parametric maps that clearly demonstrate areas of significant differences or correlations among subjects. In this last phase, while performing the statistical analysis, we design a two-sample t-test with the first group corresponding to AD patients. To obtain higher precisions in our statistical analysis, a threshold of zero adjacent voxels is applied in the two-sample comparison. The SPM8 software parameters are set as also suggested in a previous effort [11]. Figure 3 illustrates the selected clusters by the VBM analysis for one sample subject with mild AD and one sample subject affected with moderate AD.

After taking all the above steps, we have collected the clusters detected by the VBM that are required for the feature selection in the classification procedure. These detected clusters are then applied to the GM density volumes which are the results of the segmentation step of the above procedure. These clusters are actually considered as masks to specify the voxel positions. To obtain the final desired feature vectors, all the GM segmentation values for the voxel positions which are included in each one of the detected clusters, are computed. These values are then ordered in very high dimensional vectors according to the coordinate lexicographical ordering. We have now achieved our main purpose of performing this analysis which was to obtain the feature vectors containing highly important and beneficial features for our classification task. Although these vectors contain high-level features, which have the properties we are interested in for our classification model, they are very high-dimensional and quite costly to use. This is the primary reason which leads us to reduce the dimension of the vectors. We will discuss this process in detail in the following section.

#### 3.2.5. Dimension Reduction Using Principal Component Analysis (PCA)

PCA [13] is one of the best and most used tools for data representation in the least square sense for classical recognition. Commonly it is applied to decrease the dimensionality of images and still get almost all the important information embedded in the images. While performing PCA, the main focus is on finding an orthonormal set of axes which point at the direction of maximum covariance in the data. The solution is to extract the orthonormal basis vectors that are the eigenvectors of the covariance matrix of a set of images where each image is treated as a single point in a high-dimensional space. The most significant and distinguished variations between images are then mapped with these vectors. When the eigenvalues and eigenvectors of the covariance matrix are calculated, the most effective components can be chosen to form the new feature vectors with a much lower dimension. PCA is a very powerful and reliable tool for data analysis. As explained above, once the specific pattern in the data is found, they can be compressed into lower dimensions with us being confident that no valuable information will be lost.

Now that we have found and formed these very significant and beneficial feature vectors, they can be exposed to our model for a careful classification of images. Figure 4 is an illustration of the reduced feature vectors lying in the new low-dimensional space.

#### 3.2.6. Label Propagation

After taking the very fundamental step of selecting and extracting the required feature vectors, we can now continue on building up our model to reach the ultimate goal of labeling each one of the images as accurately and carefully as possible. Here, we will demonstrate the proposed approach for performing the classification in detail.

First, let us assume we have *n* different images in our dataset meaning we have extracted *n* different feature vectors each corresponding to an image. Let us assume that the number of training data items in the study equals *l* meaning we only know the labels of *l* images. Following the previously proposed method [14], first we define an n×n matrix Y with the first *l* rows corresponding to the labeled data and each column corresponding to one of the classes. One should notice that in the case of our current work, which is a classification problem with c(c=2) classes, the matrix can be defined as an n×c matrix causing no problem in the overall procedure. In a more general case this method can work with up to *n* different classes in a dataset of *n* subjects and that is why we have used Y as an n×n matrix. Also notice that here, the rest of the columns in matrix Y will not affect our results or be used or considered as a part of the required answer to the problem. This implies that the proposed method can easily be applied to any arbitrary dataset with any number of classes. In this study our main purpose is to classify the images which belong to two classes of very mild to mild AD and healthy subjects since this is the most crucial case for an efficient diagnosis of AD in order to prevent the patient’s condition from getting worse and more severe. Next, in matrix Y for every row *i*, where 1≤i≤l, we place 1 in the column corresponding to the class of $i^{th}$ labeled data and the rest of the elements will be zero. In fact, this matrix indicates the probability of each data belonging to each of the existing classes in the dataset. Next, we continue with creating matrix T as:
(6)$$T_{ij}=\frac{w_{ij}}{\sum_{k=1}^{n}w_{kj}}$$
where $w_{ij}$ is defined in Equation (Equation 5). Hence, replacing Equation (Equation 5) in Equation (Equation 6), we obtain the final definition of T:
(7)$$T_{ij}=\frac{e^{\frac{{-\left|\right|v_{i}-v_{j}\left|\right|}^{2}}{2\sigma^{2}}}}{\sum_{k=1}^{n}e^{\frac{{-\left|\right|v_{k}-v_{j}\left|\right|}^{2}}{2\sigma^{2}}}}$$

We still need to exactly determine the process of choosing the adequate value for parameter σ. As previously mentioned in Section 2.4, we use cross validation for this cause. First, a rational range of (0, 10) is chosen for σ to perform a 6-fold cross validation. Next, a set of 30 subjects is chosen for this purpose and then divided into 6 groups of 5. In each step one of these groups is selected as the validation data and the remaining 25 will be used as the training data. Finally, the best σ is chosen for the best performance through this procedure.

Now that we have specifically described matrix T, we need to follow the following steps:
Construct matrix Y and repeat the next three steps until Y converges.Replace matrix Y with TY.Normalize the rows of Y so that the sum of each row equals 1.In the end of each iteration, update matrix Y such that for every row *i*, where 1≤i≤l, replace 1 in the column corresponding to the class of $i^{th}$ labeled data and the rest of the elements in these rows will be equal to zero.

Eventually, in each row of matrix Y, the element with maximum value defines the class of the data.

Now if we consider graph *G*, which was explained in Section 2.3 and defined in Equation (Equation 5), and then normalize the weights of all existing edges for each node, we obtain matrix T. T is indeed the transition matrix of the created graph. Now the labels are in fact spreading randomly on the graph with T as the transition matrix considering that after each step all the labels on nodes are normalized and the labels of the *l* labeled training data are reset to the initial state. Figure 5 represents the different stages of this process.

As proved earlier by Xiaojin et al. [14], Y in the explained algorithm will definitely converge to a specific value. Let $Y_{L}$ and $Y_{U}$ indicate the first *l* rows and the remaining rows of Y, respectively, and let T to be written as:
(8)$$T=\left[\begin{array}{cc} T_{\mathrm{LL}} & T_{\mathrm{LU}}\\ T_{\mathrm{UL}} & T_{\mathrm{UU}} \end{array}\right]$$
where $T_{\mathrm{LL}}$ indicates a fraction of T which includes the first *l* rows and columns of it. Then, it can be proved that $Y_{U}$, which is in fact the required label matrix, is obtained from the following equation:
(9)$$Y_{U}={\left(1-T_{\mathrm{UU}}\right)}^{-1}T_{\mathrm{UL}}Y_{L}$$

## 4. Results and Discussion

In this section, we conduct experiments on the OASIS dataset to assess the effectiveness of our classification model. To understand how effective our method is in general, we conduct various experiments on the two-class subset of the dataset which contains images from MCI and NC subjects as described before. We carry out a semi-supervised learning method which requires only a small percentage of the dataset as the training data to accurately predict the labels for the remaining test data. This fact itself can illustrate the worthiness of the proposed method. We also compare the accuracy of our method against various existing approaches.

### 4.1. Competing Methods

A great amount of research has been carried out for the accurate diagnosis of cognitive diseases such as Alzheimer’s in recent years, and different approaches have been proposed for this purpose. Mostly, the information extracted from structural and functional brain imaging data or the cerebrospinal fluid is utilized for a better diagnosis. Moreover, a number of efforts have been made for the classification and prediction of different stages of AD recently. In the following, some of the most competitive works that have been carried out in this area in recent years are described:

Hosseini-Asl et al. [10]: The method proposed in this paper is basically based on a 3D convolutional auto-encoder. This is a model which applies deep 3D convolutional neural network to extract AD-related features and learn from them. Finally, the classification task is done for different binary combinations of three groups of subjects (AD, MCI, and NC) as well as a ternary classification among them.

Zu et al. [42]: In this effort, a learning method for multimodal classification of AD/MCI is represented. First feature selection is done using multiple modalities and then, utilizing a group sparsity regularizer, the different sets of extracted features are all jointly considered for selection of one subset of features which are the most informative AD-related ones. Finally to complete the classification task and for obtaining a compatible multi-task feature selection objective function, a new label-aligned regularization term is added to it. In the final step, SVM is used for mixing various feature vectors captured from multi-modality data.

Moradi et al. [43]: In this method, to create a new biomarker of MCI to AD conversion, a semi-supervised learning method is applied. While performing feature selection via regularized logistic regression on the MRI images, the aging effects are removed. Finally, for the ultimate classification which is carried out by utilizing a random forest classifier, the constructed biomarker is unified with age and cognitive measures about the MCI subjects using a supervised learning method.

Liu et al. [5]: Here, a deep learning based framework is represented for the classification of different stages of AD. In the feature selection step, stacked auto-encoders are used and since multiple neuroimaging modalities are considered. A zero-masking strategy is then applied for capturing the most discriminative features among the different modalities and the synergy between them.

Suk et al. [3]: This paper also applies deep learning for a high-level feature extraction. Deep Boltzmann Machine (DBM) is applied for this cause on a volumetric patch and is followed by another method designed for combining feature representations from different modalities. Finally, an attempt is done for solving three binary classification problems of AD/NC, MCI/NC, and MCI converter/MCI non-converter.

Casanova et al. [44]: In this paper, a new metric called AD Pattern Similarity (AD-PS) is introduced and then tested on the dataset to compare the results with the performance of the classifications which use other metrics such as the Spatial Pattern of Abnormalities for Recognition of Early AD (SPARE-AD) index. After obtaining the results from a classifier based on MRI images and another one which is trained based on cognitive measures, Casanova et al. combined the two outputs and evaluated the performance.

Chyzhyk et al. [45]: In this effort, Lattice Independent Component Analysis (LICA) is utilized for the feature selection stage as well as the Kernel transformation of the data. This approach has improved the generalization of dendritic computing classifiers. Then, the method was applied on MRI images for classification of AD patients and normal subjects.

Coupé et al. [46]: Here, the proposed method attempts to detect Alzheimer’s disease by distinguishing between specific atrophic patterns of anatomical structures such as the hippocampus (HC) and entorhinal cortex (EC). Coupé et al. attempted to capture AD-related anatomical conversions by performing segmentation and also grading of structures altogether.

Cho et al. [47]: Cho et al. represent a method for AD classification using cortical thickness data. The cortical thickness data of a subject are represented in terms of their spatial frequency components. To prevent the disruptive effects of any possible existing noise, high frequency components are filtered out. All of these help to perform an individual subject classification based on incremental learning.

Cheng et al. [48]: This paper demonstrates a domain-transfer learning method for diagnosis of AD in its different stages. The cross-domain kernel learning and then SVM are utilized to transfer supplementary domain knowledge and then perform cross-domain and auxiliary domain knowledge fusion, respectively.

Savio et al. [49]: In this effort, after obtaining the displacement vectors using non-linear registration procedures, the magnitude of the displacement vector and the Jacobian determinant of the displacement gradient matrix are extracted. Relying on the relations between these extracted values, the feature selection process is carried out. Eventually, SVM is used to reach the goal of classifying the MRI images.

Westman et al. [50]: This study aims to compare and combine MRI data from two major study cohorts in the world. After designing an automated framework for segmentation, regional volume and regional cortical thickness scores are computed and then utilized while performing multivariate analysis. In the next step, orthogonal partial least squares to latent structures (OPLS) models are created and applied to both the individual cohorts and the combined cohort for distinguishing between AD patients and healthy subjects.

Chyzhyk et al. [51]: This paper uses dendritic computing to build up a binary classifier which can also be extended to multiple classes. A single neuron lattice model with dendrite computation (SNLDC) computes an approximation of the data distribution and for a better performance, the size of the created hyperboxes are reduced. The feature extraction process is done using VBM.

Savio et al. [32]: In this paper, after applying the VBM method for extracting feature vectors from the GM segmentation volumes, different models of artificial neural networks (ANN) have been used such as: backpropagation (BP), radial basis networks (RBF), learning vector quantization networks (LVQ) and probabilistic neural networks (PNN) to perform a MCI/NC classification on brain MRI images and the best reported results were obtained with LVQ.

Chupin et al. [52]: Since hippocampal MRI volumetry (an informative biomarker for AD) has limitations due to manual segmentation, Chupin et al. introduced a fully automatic method for hippocampus segmentation. They applied probabilistic and anatomical priors for this cause. Finally, took advantage of the obtained hippocampal volumes to classify the data into three groups of AD, MCI and NC subjects.

García-Sebastián et al. [33]: In this paper, for the computation of feature vectors, VBM is applied to study the usage of both original MRI volumes and GM segmentation volumes. The SVM algorithm was applied to perform classification on the dataset consisting of patients with mild Alzheimer’s disease and control subjects.

Savio et al. [34]: This study attempted to obtain results of an Adaboost approach to AD detection in MRI brain images. Using the VBM analysis, clusters for voxel location detection are obtained and then applied to select the voxel values which lead to computation of the classification features. Next, an SVM was built upon these feature vectors. Finally, by considering various combinations of isolated classifiers, an Adaboost strategy was applied to the created SVM.

### 4.2. Parameter Tunning

In this section, the parameters of the proposed method are tuned and the evaluation procedure is described. After extracting the high dimensional features, a dimension reduction was performed using PCA. This procedure led us to choose the first 35 dimensions, which contained more than 99% of the cumulative energy. Next, we randomly chose 25 subjects to form the training set and the rest of the subjects were used as the test set. Then, in order to determine the most efficient value for σ, we performed a 6-fold cross-validation on the training set which led us to choose σ=0.25 as the best value for this parameter.

Classification accuracy, sensitivity and specificity were evaluated for different randomly chosen sets of training data with size 25, and then the values of the three metrics over 40 different runs were averaged. The results were then compared to the chosen previously proposed methods to prove outperformance of the proposed method.

### 4.3. Results

In this section, various experiments are carried out to evaluate the performance of our method. In Table 3, the performance of all existing methods is reported against the proposed method in an MCI/NC classification. All accuracy, specificity, and sensitivity scores are reported as available.

For a fair comparison, the best results of all baseline methods have been reported which clearly affirm that our proposed method has an overall better performance than other previous efforts. The accuracy and specificity of our model are by far better than the rest of the approaches. While the sensitivity score is slightly lower than that of Suk et al. [3], the accuracy of our method still clearly outperforms the previous effort [3]. Also, in Table 3 it is suggested that among all previously existing methods, Hosseini et al. [10] achieved the highest accuracy while classifying the MRI images into two classes of MCI and NC.

Considering the fact that we are proposing a semi-supervised method which only requires a small percentage of the data for training, compared to the other supervised methods, it can be understood how effective and valuable this approach can actually be for an accurate diagnosis and binary MCI/NC classification.

Table 4 represents the accuracy for different sizes of feature vectors which proves the effectiveness of the performed PCA. It illustrates that the accuracy score is increasing as the dimension increases until dim=35 and after that, the performance of our classifier is almost steady with 35 dimensions giving the best possible results.

Next, to evaluate our method’s robustness over different values of σ, we have reported the accuracy scores for a number of chosen values for this parameter in Figure 6a. From this figure it is seen that in a logical range for σ, which can be easily obtained through a k-fold cross validation procedure, our method can consistently achieve high performance results (accuracy score above 80%) where the best performance is obtained for σ=0.25. Figure 6b also demonstrates that there is an increasing trend in the performance of the classifier as the training set becomes larger, though, in a training set larger than 30, there is no significant improvement in the performance; Hence, we will not sacrifice the benefits of having a semi-supervised classification method by utilizing large groups of training data. We assume that when using computer based approaches for diagnosis, only a small portion of labeled data is available. Therefore, we tried to keep the ultimately chosen number of training data under 40% of the size of dataset.

## 5. Conclusions

In this paper, we proposed a general framework based on semi-supervised manifold learning to categorize brain MRI records in two groups of mild Alzheimer’s and normal condition (MCI/NC) with high accuracy. For distinguishing early stages of AD, we exploited a label propagation approach for the first time. We used the extracted discriminative voxel-based morphometry (VBM) features that contain the most crucial information we need. We first constructed a weighted graph based on the Euclidean distance between feature vectors. By knowing which class (MCI or NC) each of the training subjects belong to, we assigned the corresponding label to them. Then, by applying the label propagation method, we obtained the whole set of labels from just a few existing ones. We empirically demonstrated the effectiveness of our method through extensive comparison with a large group of existing methods in terms of accuracy, sensitivity, and specificity.

## Figures and Tables

**Figure 1 brainsci-07-00109-f001:**
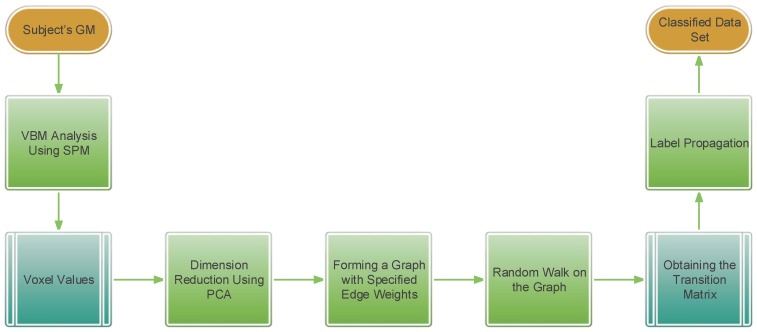
Block diagram of the proposed method. PCA: principal component analysis; VBM: voxel-based morphometry; SPM: statistical parametric map; GM: gray matter.

**Figure 2 brainsci-07-00109-f002:**
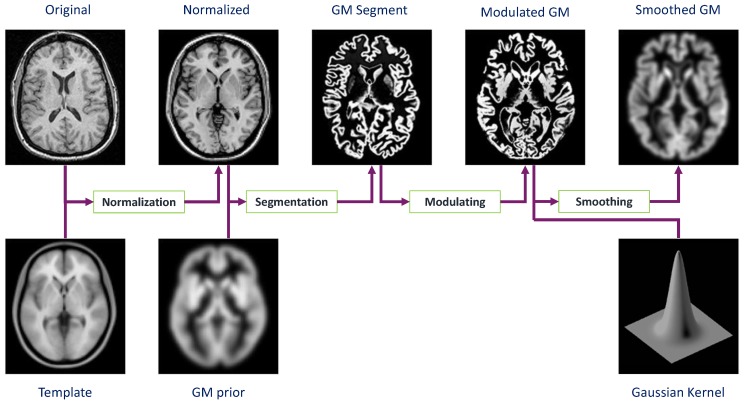
Voxel-based morphometry pre-processing overview.

**Figure 3 brainsci-07-00109-f003:**
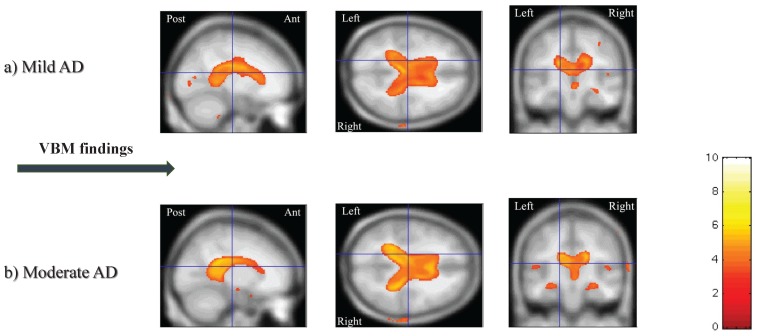
Statistical parametric maps for a subject with (**a**) mild AD and (**b**) moderate AD. The overlays show the selected clusters of features and are displayed on a sample-averaged magnetization-prepared rapid gradient echo (MP-RAGE) image on sagittal, coronal and axial sections. The color overlays show regions of statistically significant (*p*-value < 0.05) differences in rates of change compared to controls.

**Figure 4 brainsci-07-00109-f004:**
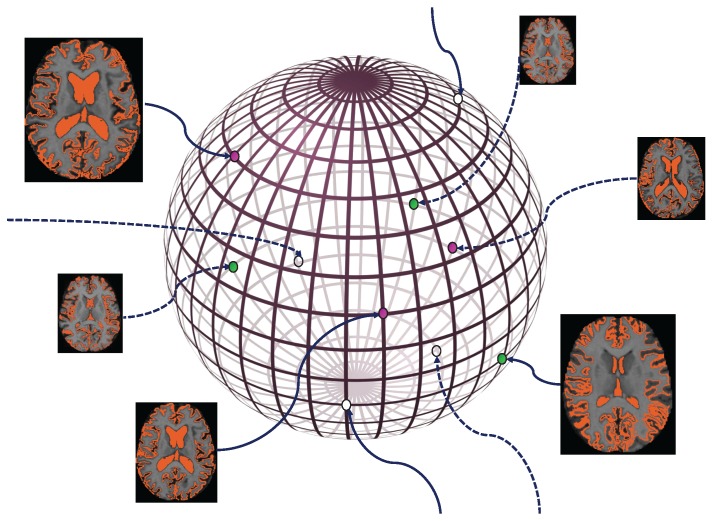
Presenting the extracted low-dimensional feature vectors from MRI images.

**Figure 5 brainsci-07-00109-f005:**
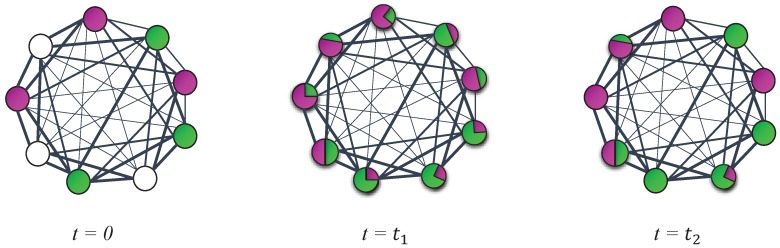
Different steps of label propagation in a fully connected graph with different edge weights which are represented with different edge widths. Each one of the green and purple colors represents the label corresponding to one of the existing classes in the dataset. The white color indicates the data being unlabeled.

**Figure 6 brainsci-07-00109-f006:**
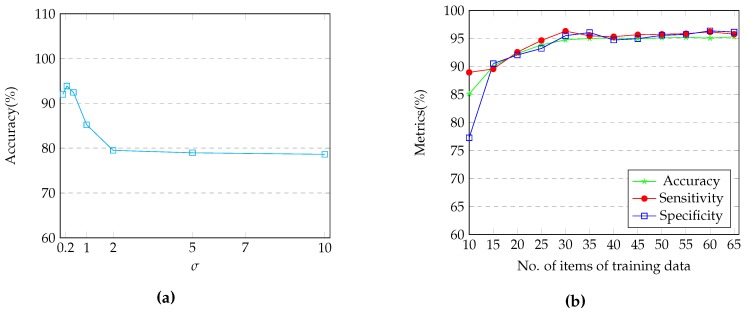
Illustrating a) performance of the proposed method over different numbers of items of training data and b) classification accuracy using the proposed method over different values of σ.

**Table 1 brainsci-07-00109-t001:** Magnetic resonance imaging (MRI) acquisition details

Sequence	MP-RAGE
TR (ms)	9.7
TE ( ms)	4
Flip Angle (°)	10
TI (ms)	20
TD (ms)	200
Orientation	Sagittal
Thickness, gap (mm)	1.25, 0
Slice No.	128
Resolution	256 × 256

ms: milliseconds

**Table 2 brainsci-07-00109-t002:** Summary of subject demographics and dementia status.

Condition	No.	Gender	Education	Socioeconomic Status	Age		CDR				MMSE	
					Range	Mean	0	0.5	1	2	Range	Mean
Very mild to mild AD	49	Both	2.63	2.94	66–96	78.08	0	31	17	1	15–30	24
Normal condition	49	Both	2.87	2.88	65–94	77.77	49	0	0	0	26–30	28.96

AD: Alzheimer’s disease; Levels of education are described as 1: Less than high school; 2: High school graduate; 3: Some college; 4: College graduate; 5: Beyond college. Categories of socioeconomic status are from 1 (highest status) to 5 (lowest status); MMSE (Mini-Mental State Examination) score ranges from 0 (worst) to 30 (best); CDR is a dementia staging instrument which gives ratings to different subjects for impairment in one of the discussed six categories.

**Table 3 brainsci-07-00109-t003:** Comparative performance (ACC, SPE, SEN %) of our MCI/NC classifier vs. other methods.

Approach	Year	Dataset	Modalities	Validation Method	Metric		
					Accuracy (%)	Sensitivity (%)	Specificity (%)
Our Method	2017	OASIS	MRI	semi-supervised method using 25% of the whole data set as training data ★	93.86	94.65	93.22
Hosseini-Asl et al. [10]	2016	ADNI	MRI	10-fold cross-validation	90.8	n/a	n/a
Zu et al. [42]	2016	ADNI	PET+MRI	10-fold cross-validation	80.26	84.95	70.77
Moradi et al. [43]	2015	ADNI	MRI	10-fold cross-validation	82	87	74
Liu et al. [5]	2015	ADNI	MRI	10-fold cross-validation	71.98	49.52	84.31
Suk et al. [3]	2014	ADNI	PET+MRI	10-fold cross-validation	85.7	99.58	53.79
Casanova et al. [44]	2013	ADNI	Only cognitive measures	10-fold cross-validation	65	58	70
Chyzhyk et al. [45]	2012	OASIS	MRI	10-fold cross-validation	74.25	96	52.5
Coupé et al. [46]	2012	ADNI	MRI	Leave-one-out cross-validation	74	73	74
Cho et al. [47]	2012	ADNI	MRI	Independent test set	71	63	76
Cheng et al. [48]	2012	ADNI	MRI	10-fold cross-validation	69.4	64.3	73.5
Savio et al. [49]	2011	OASIS	MRI	10-fold cross-validation	84	90	77
Westman et al. [50]	2011	ADNI	MRI	10-fold cross-validation	59	74	56
Chyzhyk et al. [51]	2011	OASIS	MRI	10-fold cross-validation	69	81	56
Savio et al. [32]	2009	OASIS	MRI	10-fold cross-validation	83	74	92
Chupin et al. [52]	2009	ADNI	MRI	Independent test set	64	60	65
García-Sebastián et al. [33]	2009	OASIS	MRI	Independent test set	80.61	89	75
Savio et al. [34]	2009	OASIS	MRI	10-fold cross-validation	85	78	92

★ All the existing methods use supervised learning while our proposed model utilizes a semi-supervised learning method which can further justify its efficiency. ACC: Accuracy, SPE: Specificity, SEN: Sensitivity, PET: Positron Emission Tomography, n/a: Not Available, MCI: mild cognitive impairment; NC: normal condition.

**Table 4 brainsci-07-00109-t004:** Classification accuracy using the proposed method over different feature vector sizes.

Feature vector size	10	15	20	25	30	35	40	45	50	100	200	1000
Accuracy(%)	92.33	93.15	93.37	93.42	93.75	93.86	93.84	93.75	93.77	93.70	93.63	93.77

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
