# Peer review of "Alzheimer’s Disease Early Diagnosis Using Manifold-Based Semi-Supervised Learning"

_brainsci, 2017, doi:10.3390/brainsci7080109_

Round 1
Reviewer 1 Report
Comments to Authors
The paper by Khajehnejad et al. uses voxel-based morphometric analyses of OASIS MRI brain images in a semi-supervised computational learning framework to classify the images diagnostically. The authors firstly search for discriminating, specific MRI features and extract these features characteristic of normal, MCI, or AD brains. Finally, the authors use these features for computer-based, semi-supervised learning and then use them in a label-propagation algorithm to predict the features of other sets of brain images that they used for testing the diagnostic potential of their model. They then compare the accuracy, sensitivity, and specificity of their method with other published methods. I commend the authors for coming up with this model. I have minor comments.
It will be best for the authors to tone the text of their manuscript to a more general audience. It will be best to describe some of the technical terms in the paper to befit the needs of the audience of the journal so readers don’t have to look up every term.
It is best to discuss how this method—or similar methods that authors have cited—could be rolled out in practice, in clinics, to diagnose brain MRI images. How can authors make such studies practicable?
It will also be better that the authors remove some redundancies in their text to make reading easier. For example, use of “normal condition” cases or subjects is redundant while they can use “healthy” subjects/cases.
Table 3 needs to be reformatted to include the missing column heading and make the information easy to read. The column heading “Year” is missing, and “Validation Method” is on the column heading but it appears that the actual methods are mentioned in the table. This is confusing.
The authors are also encouraged to change their style of citations to make reading easier, for example, in line 222, “Inspired by the method proposed in [11] …” does not read well as the reader would look for missing information after “in”. The citations in brackets are not noticed as normal text while reading and readers tend to ignore them when reading fast. A better alternative would be “Inspired by the method proposed previously [11] …”
Apart from redundancies and style changes while dealing with citations, I encourage the authors to improve punctuation in their manuscript.
Reference 1 in the list is incomplete. Please review all the references for completeness.
Author Response
To the Associated Editors and Reviewers
The authors would like to thank you sincerely for reviewing the manuscript. As experts in the area, the reviewers raised decent and suggestive comments to improve the quality of this manuscript. We wish the revision of the manuscript and responses to the suggestive comments are satisfactory.
Responses to Reviewer2:
We thank the reviewer for the precious comments. Apart from the general English editing in the text, please find point-to-point responses to reviewer’s comments in the following:
1. It will be best for the authors to tone the text of their manuscript to a more general audience. It will be best to describe some of the technical terms in the paper to befit the needs of the audience of the journal so readers don’t have to look up every term
In order to tone the text to a more general audience as your advice, we tried to explain some of the more professional terms in the text and describe them in more details. Here are a few examples of these changes:
“This means to calculate the geodesic distancewhich is the number of edges in a shortest path connecting them.”
“A k dimensional manifold in an m dimensional space is a surface in that space, such that for each point on this manifoldwhich have the following property; They can be mapped to a closed region in a k dimensional linear space using a diffeomorphism, which is an invertible smooth function with a smooth inverse, that maps one differentiable manifold to another. there exists a radial neighbourhood consisting of a set of points on the manifold that using a diffeomorphism can be mapped to a closed region in a k dimensional linear space.”
Also, in a couple of cases, citations were added to further clarify some of the more professional terms used in the manuscript. We also provide an example of this kind:
“Eventually, in order to build a Statistical Parametric Map (SPM) [5], we need the computed contrast which is given by the GLM estimated regression parameters. The map is then tresholded according to the Random Field theory [6] [7].”
2. It is best to discuss how this method—or similar methods that authors have cited—could be rolled out in practice, in clinics, to diagnose brain MRI images. How can authors make such studies practicable?
To address your careful concern we have thoroughly discussed it in the modified manuscript under the Introduction section:
“Diagnosis of very early AD progression is intended to aid both researchers and clinicians to develop or test new treatments and monitor their effectiveness more easily. It is stated that AD-pathologies could be detected in MRI images up to 3 years earlier than the actual clinical diagnosis [8].Therefore, a machine learning method can be of great benefit while trying to help physicians make an accurate early diagnosis. On the other hand, the expected increasing cost of caring AD patients, the workload of radiologists and the limited number of available radiologists further demonstrate the necessity of having a computer-aided system for early, fast and precise diagnosis and also for improving quantitative evaluations [9], [10]. Furthermore, all previous efforts in the field as well as the present study, when made into a computer system, can be used as a second opinion by a physician to either verify their own diagnosis and increase its reliability or improve their final decision by getting help from the computer output in cases when they are less confident about their own diagnosis. Moreover, the possibility and benefits of practical usages of computer-aided diagnosis in clinical situations, has been also the subject of a number of studies [11]. For instance, the radiologists' performance while detecting clustered microcalcifications, which are small calcium deposits in breast soft tissue, both with and without the computer output has been observed and compared in one of these studies. the radiologists' performance while detecting clustered microcalcifications, which are small calcium deposits in breast soft tissue, both with and without the computer output. It is proved in this study that the radiologists’ performance was improved significantly when the computer output was also available. As a result of these studies, computer-aided diagnosis has recently become an important part of the routine clinical process for breast cancer detection on mammograms in the United States [12].”
3. It will also be better that the authors remove some redundancies in their text to make reading easier. For example, use of “normal condition” cases or subjects is redundant while they can use “healthy” subjects/cases.
Thank you for the valuable comment. In the revised version we have omitted these redundancies and tried to rephrase them more accurately throughout the manuscript. A few examples of these changes are also reported below:
“The features must capture the most discriminativeproperties featuresthat vary between ahealthynormal conditionand a patient brain.”
“Moreover, some of the symptoms found in the AD imaging data are also captured in imaging data of aginghealthy normal condition (NC)people (age ≥75).”
“In order to discovering voxel clusters which aid us to distinguish between AD patients andhealthy normal conditionsubjects, the Statistical Parametric Mapping software, SPM8, was used to compute VBM. “
4. Table 3 needs to be reformatted to include the missing column heading and make the information easy to read. The column heading “Year” is missing, and “Validation Method” is on the column heading but it appears that the actual methods are mentioned in the table. This is confusing.
Thanks for pointing out the inaccuracy. In the new version we have corrected this mistake and Table 3 is now reformed to avoid any further confusions.
5. The authors are also encouraged to change their style of citations to make reading easier, for example, in line 222, “Inspired by the method proposed in [11] …” does not read well as the reader would look for missing information after “in”. The citations in brackets are not noticed as normal text while reading and readers tend to ignore them when reading fast. A better alternative would be “Inspired by the method proposed previously [11] …”
We have respectfully considered your comment and changed the style of our citations in text, therefore there are no such misleading sentences in the revised manuscript anymore.
6. Apart from redundancies and style changes while dealing with citations, I encourage the authors to improve punctuation in their manuscript
Thanks to your comments on the matter of punctuation in the manuscript, we have reviewed the manuscript once more and enhanced this issue while also trying to check the English language and making improvements by editing the text.
7. Reference 1 in the list is incomplete. Please review all the references for completeness.
Reference 1 is now complete and the whole reference list is double checked to avoid any other incompleteness.
References:
[1] Friston, K.J.; Holmes, A.P.; Worsley, K.J.; Poline, J.P.; Frith, C.D.; Frackowiak, R.S. Statistical parametric maps in functional imaging: a general linear approach. Human brain mapping 1994, 2, 189–210.
[2] Brett, M.; Penny, W.; Kiebel, S. Introduction to random field theory. Human brain function 2003, 2.
[3] J. Cao and K. Worsley, "Applications of random fields in human brain mapping, LECTURE NOTES IN STATISTICS-NEW YORK-SPRINGER VERLAG- 2001, pp. 169–182.
[4] Adaszewski, S.; Dukart, J.; Kherif, F.; Frackowiak, R.; Draganski, B.; Initiative, A.D.N.; others. How early can we predict Alzheimer’s disease using computational anatomy? Neurobiology of aging 2013, 34, 2815–2826.
[5] Bron, E.E.; Smits,M.; Van Der Flier,W.M.; Vrenken, H.; Barkhof, F.; Scheltens, P.; Papma, J.M.; Steketee, R.M.; Orellana, C.M.; Meijboom, R.; others. Standardized evaluation of algorithms for computer-aided diagnosis of dementia based on structural MRI: the CADDementia challenge. NeuroImage 2015, 111, 562–579.
[6] van Ginneken, B.; Schaefer-Prokop, C.M.; Prokop, M. Computer-aided diagnosis: how to move from the laboratory to the clinic. Radiology 2011, 261, 719–732.
[7] Doi, K. Computer-aided diagnosis inmedical imaging: historical review, current status and future potential. Computerized medical imaging and graphics 2007, 31, 198–211.
[8] Doi, K. Diagnostic imaging over the last 50 years: research and development in medical imaging science and technology. Physics in medicine and biology 2006, 51, R5.

Reviewer 2 Report
This paper proposes a manifold based semi-supervised learning method for Alzheimer’s disease diagnosis at early symptomatic stages. The framework includes the following steps. First, voxel based morphometry is applied to extract most critical AD-related features in gray matter of each subject. Second, PCA is used to reduce the dimension of the feature vectors. Third, a graph is constructed from the feature vectors where each feature vector represents a point in a k-dimension manifold. Finally, given a subset of subjects whose group labels are known, the random walk based label propagation method is utilized to categorize unlabeled subjects. The novelty of the proposed method lies in steps 3 and 4. It was tested in 96 subjects from the Open Access Series of Imaging Studies, OASIS, where 49 of whom are very mild to mild AD patients. The authors conducted thorough experiments to evaluate the effectiveness and efficiency of the proposed method and compared it with many recent studies.
I have the following comments and concerns.
1. Section 2 is really huge, it is better to move 2.1-2.4 to another section called “Theoretical Background”.
2. The authors stated that they chose 96 subjects from OASIS as these subjects have complete demographic, clinical or derived anatomic volumes information. It is hardly believable that only 96 out of 416 subjects have this complete set of information in a famous database like OASIS.
3. It was stated that “three to four T1-weighted magnetization-prepared rapid gradient echo images were captured from every subject”. In this paper, how many images were used for each subject and how these images were selected?
4. It is better to put ADNI as a proposed future work rather than explaining why it was not used in the current study, as ADNI has far more subjects and far more widely spanned age range than OASIS, even for the normal control group.
5. In line 157, it is stated “There are two different approaches to solve this type of classification problems”, but only one approach was talked.
6. Section 2.4.1 is not clearly written, please elaborate.
7. Please add a figure to illustrate the selected cluster of features overlaid on a gray matter image, at lease for one slice.
8. In line 301, “first we define an n x n matrix Y with the first l rows corresponding to the labeled data and each column corresponding to one of the classes”, shouldn’t it be “an n x c matrix”, where c is the number of classes?
9. Please consider putting Sec. 2.7 as Sec. 2.6.1 to give readers a better understanding of the overall framework.
Author Response
To the Associated Editors and Reviewers
The authors would like to thank you sincerely for reviewing the manuscript. As experts in the area, the reviewers raised decent and suggestive comments to improve the quality of this manuscript. We wish the revision of the manuscript and responses to the suggestive comments are satisfactory.
Responses to Reviewer1:
We thank the reviewer for the precious comments. Apart from the general English editing in the text, please find point-to-point responses to reviewer’s comments in the following:
1. Section 2 is really huge, it is better to move 2.1-2.4 to another section called “Theoretical Background”.
The required change was applied in the manuscript dividing the previous section 2 in two different sections named “Theoretical Backgrounds” and “Methods and Materials”.
2. The authors stated that they chose 96 subjects from OASIS as these subjects have complete demographic, clinical or derived anatomic volumes information. It is hardly believable that only 96 out of 416 subjects have this complete set of information in a famous database like OASIS.
Thank you for raising this question. Since we were aiming to evaluate our method under more challenging conditions where subjects were chosen from the older ones (65-96 years old), we had to choose subjects which met both conditions of having all the required information needed for the usage of SPM8 software and the VBM analysis, and also being in the desired range of age. Also, we were aiming to have the same number of subjects in both classes. Therefore, it actually is true that only 98 subjects (49 from each class) had these requirements. This choice of the exploited dataset has been also used in other previous efforts [1], [2], [3] which are now mentioned in the revised manuscript as well.
3. It was stated that “three to four T1-weighted magnetization-prepared rapid gradient echo images were captured from every subject”. In this paper, how many images were used for each subject and how these images were selected?
Thank you for pointing out this matter. In “Open Access Series of Imaging Studies (OASIS): Cross-sectional MRI Data in Young, Middle Aged, Nondemented, and Demented Older Adults” by Marcus et al. [4], it is mentioned that “For registration, a 12-parameter affine transformation was computed to minimize the variance between the first MP-RAGE image and the atlas target. The remaining MP-RAGE images were registered to the first (in-plane stretch allowed) and resampled via transform composition into a 1-mm isotropic image in atlas space. The result was a single, high-contrast, averaged MP-RAGE image in atlas space. Subsequent steps included skull removal by application of a loose-fitting atlas mask and correction for intensity inhomogeneity due to nonuniformity in the magnetic field. Intensity variation was corrected across contiguous regions, based on a quadratic inhomogeneity model fitted to data from a phantom” which clarifies that in the OASIS dataset we have one single averaged image for each subject.
A paragraph explaining this matter is also added to the manuscript to clarify any misunderstanding that could have occurred in the previous version:
“In this study, we will exploit the averaged MP-RAGE image for each subject which is obtained through registration. First, for minimizing the variance between the first MP-RAGE image and the atlas target, which has been described in detail by Marcus et al. [4], a 12-parameter affine transformation was computed. Then a single, high-contrast, averaged MP-RAGE image was produced in atlas space by registering the remaining MP-RAGE images to the first one (in-plane stretch allowed) and resampling via transform composition into a 1-mm isotropic image in atlas space. This process is discussed in more detail previously [4].”
4. It is better to put ADNI as a proposed future work rather than explaining why it was not used in the current study, as ADNI has far more subjects and far more widely spanned age range than OASIS, even for the normal control group.
Following your valuable comment, the paragraph discussing why OASIS dataset had been chosen instead of ADNI, was omitted and instead we will consider applying our method to the ADNI dataset in our future work. This is also added to the manuscript:
“As our future work, we aim to apply our method on the other well-known data base: Alzheimer’s Disease Neuroimaging Initiative (ADNI) as well.”
5. In line 157, it is stated “There are two different approaches to solve this type of classification problems”, but only one approach was talked.
Thank you very much for your careful reviews. This part had been omitted due to a mistake and we have added it back to section 2.4 which explains the second approach.
“In the second approach, according to the probabilities of each data belonging to each class in the corresponding defined classification problem, each unlabeled data will eventually belong to the class where it had been assigned the highest value probability.”
6. Section 2.4.1 is not clearly written, please elaborate.
More explanations are added to section 2.4.1. We hope we have provided enough details on the matter to clarify it furthermore.
“… Based on this fact, a class of label propagation methods have been developed, which can be explained in more details as follows. In the first step, each labeled data acting as a labeled node in the graph, has its own label with a weight which equals to 1. Next, in each step, the labeled nodes distribute their labels among all their neighbours giving their label to each neighbouring node with a weight equal to the normalized weight of the edge between them. At the end of each step, the primarily labeled data gain back their own original label while the unlabeled data now have new sets of labels on them for continuing the process. This iterative procedure goes on until reaching a stationary state for the labels on all the nodes.”
7. Please add a figure to illustrate the selected cluster of features overlaid on a gray matter image, at lease for one slice.
A figure (Figure 3) trying to illustrate the selected clusters during VBM analysis is added to the manuscript. It shows the selected clusters of features which are displayed on sample averaged MP-RAGE images on sagittal, coronal and axial sections. These detected clusters are then applied to the GM density volumes which are the results of the segmentation step.
8. In line 301, “first we define an n×n matrix Y with the first l rows corresponding to the labeled data and each column corresponding to one of the classes”, shouldn’t it be “an n×c matrix”, where c is the number of classes?
Thank you for pointing this out. In the case of our current work, which is a classification problem with c (c=2) classes, the matrix can be defined as an n×c matrix causing no problem in the overall procedure; Though in a more general case this method can work with up to n different classes in a dataset of n subjects and that is why we have used Y as an n×n matrix. We should notice that in our current work the rest of the columns in matrix Y will not affect our results or be used or considered as a part of the required answer to the problem.
In order to clarify this matter in the manuscript, the following explanations are also added to the Label Propagation section:
“One should notice that in the case of our current work, which is a classification problem with c (c = 2) classes, the matrix can be defined as an n × c matrix causing no problem in the overall procedure; Though in a more general case this method can work with up to n different classes in a dataset of n subjects and that is why we have used Y as an n × n matrix. Also notice that here, the rest of the columns in matrix Y will not affect our results or be used or considered as a part of the required answer to the problem.”
9. Please consider putting Sec. 2.7 as Sec. 2.6.1 to give readers a better understanding of the overall framework.
The subsection was moved to the beginning of section “Methods and Materials” to give readers a better overlook and understanding of the whole procedure.
References:
[1] Savio, A.; García-Sebastián, M.; Hernández, C.; Graña, M.; Villanúa, J. Classification results of artificial neural networks for alzheimer’s disease detection. International Conference on Intelligent Data Engineering and Automated Learning. Springer, 2009, pp. 641–648.
[2] Savio, A.; García-Sebastián, M.; Graña, M.; Villanúa, J. Results of an adaboost approach on Alzheimer’s disease detection on MRI. Bioinspired Applications in Artificial and Natural Computation 2009, pp. 114–123.
[3] García-Sebastián, M.; Savio, A.; Graña, M.; Villanúa, J. On the use of morphometry based features for Alzheimer’s disease detection on MRI. International Work-Conference on Artificial Neural Networks. Springer, 2009, pp. 957–964.
[4] Marcus, D.S.; Wang, T.H.; Parker, J.; Csernansky, J.G.; Morris, J.C.; Buckner, R.L. Open Access Series of Imaging Studies (OASIS): cross-sectional MRI data in young, middle aged, nondemented, and demented older adults. Journal of cognitive neuroscience 2007, 19, 1498–1507.

Round 2
Reviewer 2 Report
The authors fully addressed my concerns. I do not have any further comments.
Author Response
Response to Reviews of Round 2:
We would like to thank the reviewers for their constructive criticism which have assisted us to make our manuscript more well-suited and qualified for this prestigious journal.
Since the responses to the first round of reviews appeared to have successfully addressed all of the reviewers’ precise and valuable comments and concerns, on this round we respectfully embarked on mainly editing the text and making revisions in the English language and style of the manuscript.
Following the comments regarding English language and style and also the required spell checking, we would like to inform you with pleasure that we have once more done a thorough review of the text and done detailed spell checking to make sure that the English language of the manuscript meets the standard of the journal. We sincerely wish that the latest changes are sufficient and satisfactory.
